# Local chromatin environment of a Polycomb target gene instructs its own epigenetic inheritance

Scott Berry, Matthew Hartley, Tjelvar S G Olsson, Caroline Dean*, Martin Howard*

John Innes Centre, Norwich, United Kingdom

**Abstract** Inheritance of gene expression states is fundamental for cells to 'remember' past events, such as environmental or developmental cues. The conserved Polycomb Repressive Complex 2 (PRC2) maintains epigenetic repression of many genes in animals and plants and modifies chromatin at its targets. Histones modified by PRC2 can be inherited through cell division. However, it remains unclear whether this inheritance can direct long-term memory of individual gene expression states (cis memory) or instead if local chromatin states are dictated by the concentrations of diffusible factors (trans memory). By monitoring the expression of two copies of the *Arabidopsis* Polycomb target gene *FLOWERING LOCUS C* (*FLC*) in the same plants, we show that one copy can be repressed while the other is active. Furthermore, this 'mixed' expression state is inherited through many cell divisions as plants develop. These data demonstrate that epigenetic memory of *FLC* expression is stored not in trans but in cis.

*For correspondence: caroline. dean@jic.ac.uk (CD); martin. howard@jic.ac.uk (MH)

Competing interests: The authors declare that no competing interests exist.

## Introduction

Epigenetic memory can be stored in the concentrations of diffusible regulatory factors that are maintained through feedback loops (trans memory) (*Novick and Weiner, 1957*; *Ptashne, 2004*; *Zacharioudakis et al., 2007*; *Xu et al., 2009*). Alternatively, memory could be stored locally in the chromatin environment of individual genes (cis memory), in the form of DNA methylation or post-translational modifications of histones (*Moazed, 2011*). While in both trans and cis memory the chromatin state is inherited, in the former case chromatin *responds* to the transcriptional state defined by heritable concentrations of the trans-factors, whereas in the latter case it is the local chromatin environment that *instructs* its own inheritance and is, therefore, the key epigenetic memory element.

Whether heritable information can be stored in patterns of histone modifications is a subject of much debate (*Ptashne, 2007*; *Kaufman and Rando, 2010*; *Henikoff and Shilatifard, 2011*; *Petruk et al., 2012*; *Gaydos et al., 2014*). Central to this debate is Polycomb repressive complex 2 (PRC2), a key transcriptional regulator in animals and plants. PRC2 catalyses methylation of histone H3 on lysine 27 (H3K27me) (*Cao et al., 2002*) and mutagenesis experiments have shown that K27 is necessary for PRC2-mediated repression (*Pengelly et al., 2013*). DNA replication poses a significant challenge to histone-modification based memory as patterns of modifications must be faithfully transmitted to daughter DNA strands (*Steffen and Ringrose, 2014*). During replication, maternal histones are shared between daughter chromosomes where they re-associate close to their original location (*Radman-Livaja et al., 2011*; *Annunziato, 2005*). Since PRC2 has affinity for H3K27me3, it has been proposed that H3K27me3 on inherited histones could recruit PRC2 to daughter DNA strands in order to similarly modify newly incorporated, unmodified histones (*Margueron et al., 2009*). Supporting the hypothesis that modified histones are indeed the key inherited cis memory element, it was recently observed that passage of H3K27-methylated histones to daughter chromosomes can occur

**eLife digest** Genetic material is contained within molecules of DNA. In plants and many other organisms, these DNA molecules are packaged around proteins called histones to make a structure known as chromatin. Altering how the DNA is packaged in chromatin can control the activity of genes. For example, a group of proteins called the Polycomb Repressive Complex 2 (PRC2) adds methyl tags to histones, which can alter the packaging of chromatin to lower the activity of particular genes.

When a cell divides, it is sometimes important that genes in the daughter cells have similar levels of activity as the parent cell. This allows individual cells to 'remember' past events, such as exposure to cold temperatures or other environmental conditions. The pattern of methyl tags on histones can be passed onto the daughter cells, but it is not clear if this is actually responsible for providing the memory.

One gene that PRC2 regulates is called *FLC*, which influences when a plant called *Arabidopsis* produces flowers. If the plants are exposed to cold temperatures, the activity of *FLC* is repressed. *FLC* activity remains low after the period of cold has ended to ensure that the plants produce flowers at an appropriate time. If this 'memory of cold' is held locally in the chromatin of the *FLC* gene, then it should be possible for two copies of the *FLC* gene in the same cell to show different gene activities. However, if the memory is stored more globally inside each cell by other proteins, then the two copies should have identical gene activities.

To distinguish between these two possibilities, Berry et al. added different fluorescent tags to two copies of the *FLC* gene in *Arabidopsis* plants, which allowed the activity of each gene copy to be tracked in individual cells under a microscope. The experiments show that one copy of *FLC* may be switched off, while the other remains switched on inside the same cell. Furthermore, it was found that this pattern of gene activity is passed onto the daughter cells when the original cell divides.

Berry et al.'s findings show that the memory of *FLC* gene activity is stored locally in the chromatin of the *FLC* gene itself. The alteration of histones by PRC2 is one important aspect of the packaging of chromatin. The next challenge is to identify what other features of chromatin are required for a gene to be able to store this memory locally.

in the absence of functional PRC2 (*Gaydos et al., 2014*). Other studies have suggested that Polycomb proteins themselves are maintained locally through DNA replication (*Petruk et al., 2012*; *Francis et al., 2009*).

These studies provide valuable mechanistic information about how a Polycomb system *could* store the memory of transcriptional states of individual genes in cis. However, to definitively demonstrate that memory *is* stored in cis, it is necessary to show that two copies of the same DNA sequence can be independently maintained in different transcriptional states in the same nucleus (*Bonasio et al., 2010*). This has been observed in genomic imprinting and in random X-chromosome inactivation. Since imprinting involves DNA methylation (*Ferguson-Smith, 2011*), and X-chromosome inactivation involves chromosome-wide changes in chromatin structure and nuclear positioning (*Gendrel et al., 2014*), it remains an open question whether Polycomb-repressed chromatin at a single gene can store epigenetic memory. Cis memory has also been implicated in random monoallelic expression. However, studies so far have been limited to genes with naturally occurring genetic polymorphisms (*Eckersley-Maslin et al., 2014*; *Deng et al., 2014*) or are performed on fixed tissues limiting conclusions regarding heritability (*Gendrel et al., 2014*). Importantly, a requirement for chromatin-modifying factors in epigenetic memory does not necessarily imply that memory is itself stored in chromatin.

To address the question of whether Polycomb-repressed chromatin at a single gene can instruct its own inheritance and therefore constitutes a cis memory system, we exploited the classic epigenetic process of vernalization in *Arabidopsis*. Vernalization is the acceleration of flowering following prolonged cold exposure and is mediated by cold-induced epigenetic repression of the Polycomb target gene and floral repressor *FLC*. *FLC* repression requires PRC2 but is independent of DNA methylation (*Finnegan et al., 2005*), making it a useful system for studying the cis-memory storage capability of Polycomb at a single gene. In vernalized plants, *FLC* expression is bistable, with the

number of cells in which *FLC* is stably repressed increasing quantitatively with the duration of prior cold exposure (*Angel et al., 2011*). Cold exposure results in localization of a plant-homeodomain-PRC2 complex (PHD-PRC2) to a small nucleation region within the *FLC* gene, which leads to locally increased H3K27me3 and co-ordinately decreased H3K36me3 (*Angel et al., 2011*; *De Lucia et al., 2008*; *Finnegan and Dennis, 2007*; *Yang et al., 2014*). On return to warm, the PHD-PRC2 complex spreads across the gene leading to high H3K27me3 over the whole locus and stable repression (*Angel et al., 2011*; *De Lucia et al., 2008*). After cold, H3K27me3 is present at the *FLC* promoter and gene body but does not spread to neighbouring genes (*Angel et al., 2011*; *Finnegan and Dennis, 2007*; *Yang et al., 2014*). Polycomb-dependent *FLC* repression is maintained throughout plant development until it is reset during embryogenesis (*Gendall et al., 2001*; *Crevillen et al., 2014*). If *FLC* transcriptional states are defined by concentrations of diffusible factors (trans memory), then all copies of *FLC* within a cell should be in the same transcriptional state. On the other hand, if *FLC* chromatin states instruct their own inheritance (cis memory), then it should be possible for different copies of *FLC* within a cell to exist in different heritable transcriptional states.

## Results and discussion

To search for evidence of this 'mixed' transcriptional state, we generated a system in which we could visualize the expression of two copies of *FLC* in single cells. We generated plants expressing FLC tagged with either Venus or mCherry fluorescent protein in the *FRI*-sf2 *flc-2* genetic background (*Michaels et al., 1999*). The transgenes consist of 12 kb of genomic DNA surrounding *FLC* with the *Venus* or *mCherry* sequence inserted as a translational fusion into *FLC* exon 6 (*Figure 1A*). The early flowering phenotype of *flc-2* mutants was fully rescued in transgenic lines, demonstrating that the fusion proteins are functional (*Figure 1B,C*). Expression of full-length FLC-Venus fusion protein was confirmed by immunoprecipitation, followed by immunoblot and mass spectrometry (*Figure 1—figure supplement 1*). Like wild-type Columbia line *FRI*-sf2 (Col-FRI), *FLC-Venus* and *FLC-mCherry* plants responded to cold exposure by accelerating flowering (*Figure 1C*). This indicates that the *FLC* transgenes were stably repressed similarly to endogenous *FLC* after cold. Epigenetic repression was confirmed to be quantitatively dependent on the duration of cold exposure for *FLC-Venus* at the transcriptional level in both roots and shoots, and also at the protein level (*Figure 2A*, *Figure 2–figure supplement 1*). We also verified that the *FLC* antisense transcripts, named *COOLAIR* (*Swiezewski et al., 2009*; *Csorba et al., 2014*), are expressed from these constructs and are induced during cold exposure (*Figure 2—figure supplement 2*).

FLC-Venus and FLC-mCherry were visualized using confocal microscopy of root meristems in warm conditions 7 days after cold exposures ranging from 2 to 10 weeks. Roots provide an excellent system for studying heritability of epigenetic states because cell lineages are visible as continuous files of cells that arise from repeated anticlinal divisions along the longitudinal axis of the growing root (*Dolan et al., 1993*) (*Figure 2B*). In warm conditions, stem cells at the root tip divide approximately once every two days, while for other meristematic cells this occurs approximately once per day (*Campilho et al., 2006*). Each cell undergoes several divisions before reaching the elongation zone whereupon cell division no longer takes place and cells begin endoreduplication. Therefore, 7 days after plants are transferred from cold, a single cell in the stem cell niche will have given rise to a lineage that encompasses a long file of cells in the root meristem. These files allow a direct assay of the mitotic stability of the epigenetic state in vivo. Strikingly, images of FLC-Venus and FLC-mCherry 7 days after cold exposure showed long files of cells in the same expression state (*Figure 2C*, *Figure 2—figure supplement 3*, *Figure 2—figure supplement 4*), demonstrating the long-term mitotic stability of the active and repressed transcriptional states. Such files make it implausible that the observed expression levels in single cells are the result of transcriptional noise. To quantitatively characterize the *FLC-Venus* expression status in individual cells after different durations of cold, we developed an automated image analysis procedure to calculate the mean FLC-Venus intensity inside each cell ('Materials and methods' and *Figure 2—figure supplement 5*, *Figure 2—figure supplement 6*). As suggested by individual images, our unbiased analysis of 53 roots (8547 reconstructed cells) revealed that the number of cells expressing *FLC-Venus* decreased quantitatively with the duration of cold exposure (*Figure 2C,D*). Importantly, *FLC-Venus* expression was bimodal after 6 or 8 weeks cold, and almost all cells were silenced after 10 weeks cold (*Figure 2D*).

Having developed an assay to visualize mitotic heritability of *FLC* expression states at the single-cell level, we returned to the question of cis vs trans epigenetic memory at *FLC*. We crossed *FLC-mCherry* to

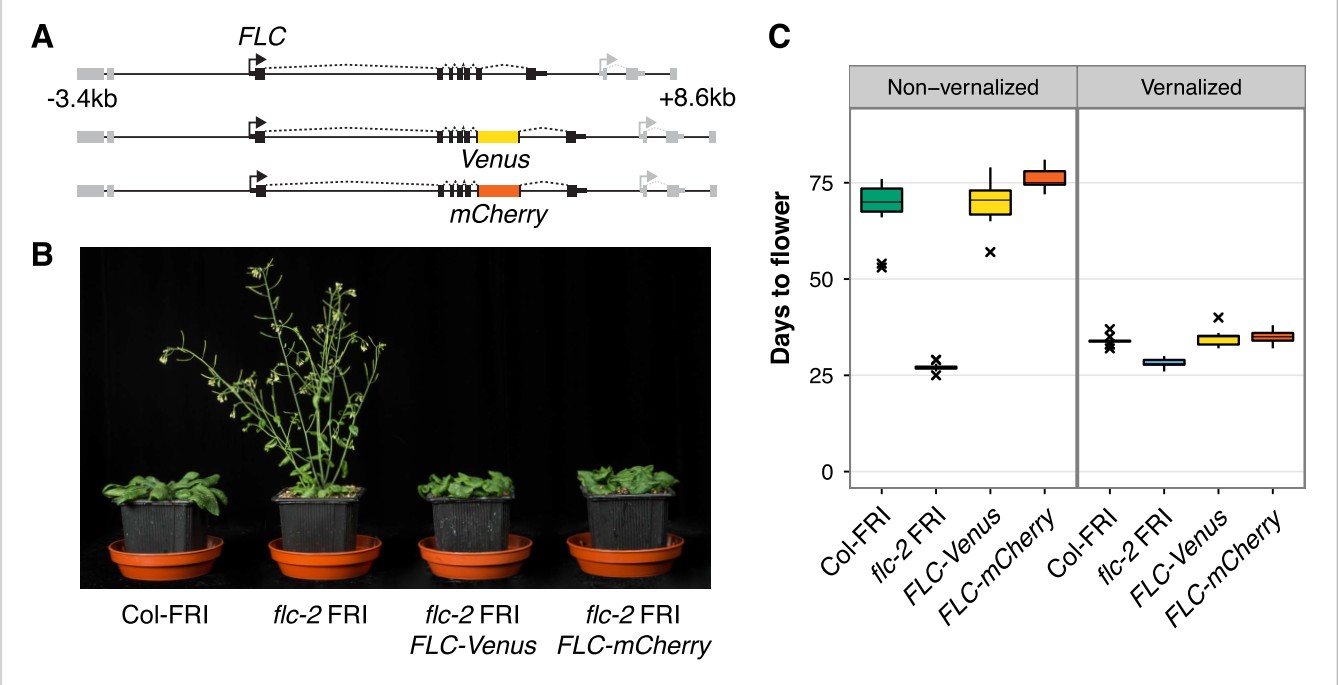

**Figure 1**. *FLC-Venus* and *FLC-mCherry* plants are late flowering and vernalization responsive. (**A**) Schematic of *FLC* genomic DNA used to generate *FLC-Venus* and *FLC-mCherry* translational fusions. Exons are represented by black boxes. Transgenes extend from 3.4 kb upstream to 8.6 kb downstream of the *FLC* transcription start site. Neighbouring genes are depicted in grey. (**B**) Photograph showing the early-flowering phenotype of non-vernalized parental *flc-2* FRI plants and the complementation of *flc-2* in transgenic *FLC-Venus* and *FLC-mCherry* plants. (**C**) Flowering time for homozygous single-copy *FLC-Venus* and *FLC-mCherry* plants (single transgenic line each) compared to wild-type (Columbia line *FRI*-sf2 [Col-FRI]) and parental *flc-2* FRI plants (n = 12); Vernalized plants were pre-grown for 1 week at 22°C and spent 4 weeks at 5°C before being returned to 22°C. Flowering time is counted in days from sowing until bolting but does not include time spent in cold.

The following figure supplement is available for figure 1:

**Figure supplement 1**. Detection of FLC-Venus protein.

*FLC-Venus* plants to generate F1 hybrids carrying a single copy of each transgene. In a trans-based memory, the only possible heritable expression states of *FLC-Venus*/*FLC-mCherry* are ON/ON and OFF/OFF, because epigenetic information is stored as a diffusible signal (***Figure 3A***). In a cis-based memory, all four states ON/ON, ON/OFF, OFF/ON, and OFF/OFF are possible because the information is stored at the locus itself (***Figure 3B***). As expected, non-vernalized roots showed uniform ON/ON expression of both transgenes in all cells (***Figure 3C***, ***Figure 3—figure supplement 1***). To stochastically induce repression of the two *FLC* transgenes, we exposed F1 plants to 4–6 weeks of cold followed by 7 days growth in warm. Strikingly, we observed long files of cells in which one *FLC* reporter was stably repressed, while the other remained stably activated (***Figure 3D***). In fact, we observed long-term mitotic stability of all four possible combinations: *FLC-Venus*/*FLC-mCherry* ON/ON, ON/OFF, OFF/ON, and OFF/OFF (***Figure 3—figure supplement 2***). Inheritance of the 'mixed' ON/OFF and OFF/ON states directly contradicts trans memory and instead provides direct evidence of a cis-encoded epigenetic state at *FLC*.

The *FLC-Venus* and *FLC-mCherry* transgenes in these double-hemizygous F1 plants are not in the same genomic location. However, like endogenous *FLC*, the reporters are actively expressed throughout development in warm conditions and epigenetically repressed specifically in response to prolonged cold (***Figures 1–3***, ***Figure 2—figure supplement 1***, ***Figure 2—figure supplement 3***, ***Figure 2—figure supplement 4***, ***Figure 3—figure supplement 1***, ***Figure 3—figure supplement 2***). Epigenetic repression, therefore, depends on cold exposure and the *FLC*-specific sequences within the transgenes rather than the genomic location. Furthermore, since we observe similar numbers of

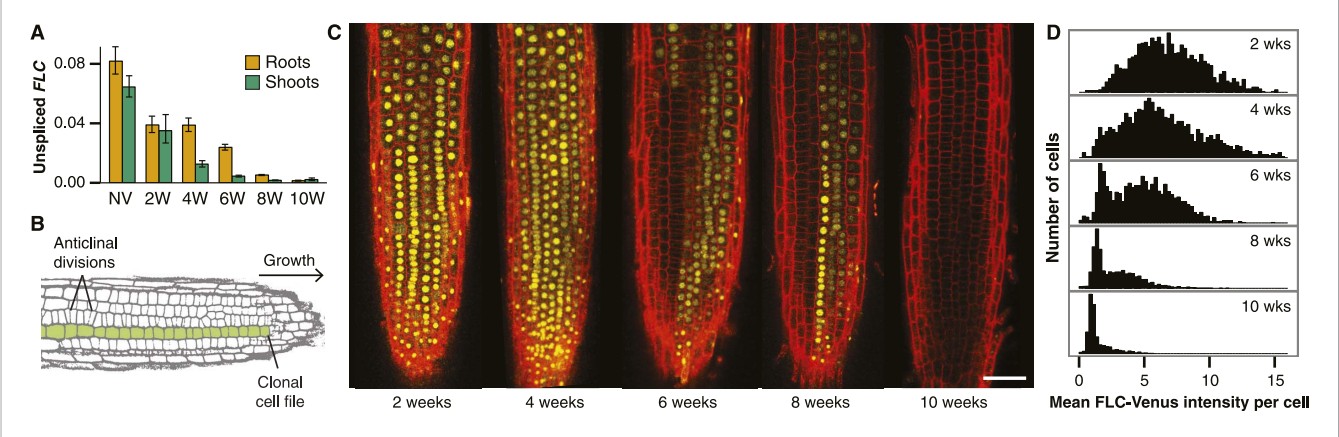

**Figure 2**. Active and repressed *FLC-Venus* transcriptional states are mitotically inherited. (**A**) Unspliced *FLC-Venus* RNA in roots and shoots measured by qRT-PCR for non-vernalized (NV) plants or after 2, 4, 6, 8, or 10 weeks of cold followed by 7 days of warm. Data shown are mean ± s.e.m. for at least 2 biological replicates for each of 2 independent transgenic lines (n ≥ 4). (**B**) Schematic of an *Arabidopsis* root meristem showing how repeated anticlinal cell divisions give rise to clonal cell files along the axis of growth. (**C**) Confocal microscopy images of FLC-Venus (yellow) in primary root meristems of plants exposed to 2, 4, 6, 8, or 10 weeks cold. Plants were imaged 7 days after return to warm. FLC-Venus is localized to nuclei. Cell walls were stained with propidium iodide (red). Scale bar, 50 μm. (**D**) Histograms of mean FLC-Venus intensity in individual cells (see 'Materials and methods') for two independent *FLC-Venus* transgenic lines. Each panel summarizes data from confocal z-stacks of 8–12 roots (1372–2067 cells).

The following figure supplements are available for figure 2:

**Figure supplement 1**. *FLC-Venus* is quantitatively epigenetically repressed by cold exposure.

**Figure supplement 2**. *COOLAIR* expression in transgenic *FLC* lines.

**Figure supplement 3**. Confocal microscopy of FLC-Venus and FLC-mCherry.

**Figure supplement 4**. Confocal microscopy of *FLC-Venus* after vernalization.

**Figure supplement 5**. Quantitative image analysis.

**Figure supplement 6**. Root-to-root variability in quantified images.

cells in the *FLC-Venus/FLC-mCherry* ON/OFF and OFF/ON expression states (*Figure 3—figure supplement 2*), we consider it unlikely that differences in genomic location or fluorophore sequence between the transgenes causes one of the copies to be preferentially repressed over the other.

Our findings demonstrate that the molecular changes to the chromatin environment of *FLC* induced by prolonged cold exposure are sufficient to instruct epigenetic inheritance of the Polycomb-repressed transcriptional state. However, Polycomb complex binding and H3K27me may not be the only factors that constitute this locally encoded epigenetic state. Specific nucleic acid structures (*Klose et al., 2013*), non-coding RNAs (*Herzog et al., 2014*) (for example the chromatin-associated *COOLAIR* antisense transcripts at *FLC* (*Csorba et al., 2014*)), and histone variants (*Jacob et al., 2014*) may also play important roles. Thus, not all H3K27-methylated chromatin should be considered as epigenetically silent. What defines a local chromatin environment as sufficient to confer cis-based epigenetic memory is now a central question in this field. Our methodology of using two reporters to specifically distinguish cis vs trans memory is broadly applicable and should be helpful in this pursuit.

## Materials and methods

### Plant materials and growth conditions

Plant growth conditions were described previously (*De Lucia et al., 2008*). For expression analysis and protein extraction, plants were grown on Murashige and Skoog (MS) agar plates without glucose.

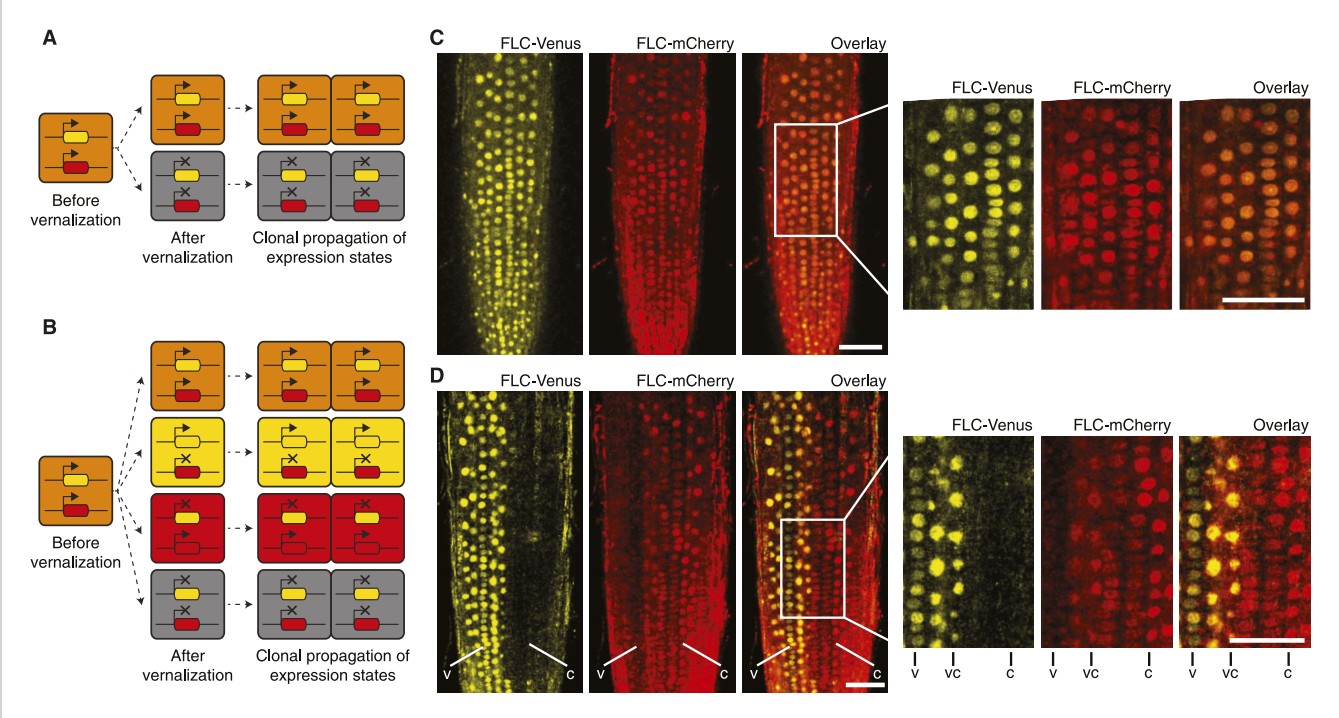

**Figure 3**. 'Mixed' transcriptional states are mitotically inherited. (**A**) In a trans memory system, the two copies of *FLC* are coordinately regulated and only two mitotically heritable states are possible (*FLC-Venus*/*FLC-mCherry* ON/ON, OFF/OFF). (**B**) In a cis memory system, the two copies of *FLC* can be maintained in alternative expression states, so four mitotically heritable states are possible (*FLC-Venus*/*FLC-mCherry* ON/ON, ON/OFF, OFF/ON, OFF/OFF). (**C**) Non-vernalized roots of *FLC-mCherry* × *FLC-Venus* F1 plants show uniform expression of *FLC-Venus* and *FLC-mCherry* in all nuclei. (**D**) After vernalization such plants can epigenetically repress a single-gene copy, while the other remains activated. The following notation is used to indicate files of cells in the various expression states: Both expressed, vc; *FLC-Venus* only, v; *FLC-mCherry* only, **C**. Scale bars, 50 μm.

The following figure supplements are available for figure 3:

**Figure supplement 1**. Confocal images of NV *FLC-Venus* × *FLC-mCherry* F1 plants.

**Figure supplement 2**. Confocal images of vernalized *FLC-Venus* × *FLC-mCherry* F1 plants.

For flowering time measurements, plants were transferred from plates to soil 5 days after vernalization. For microscopy, plants were grown almost vertically on MS plates supplemented with 1% (wt/vol) sucrose and 0.5% (wt/vol) Phytagel (Sigma–Aldrich, St. Louis, MO, P8169).

Col-FRI is Col-0 with an introgressed active *FRIGIDA* allele (*FRI*-sf2), and was previously described (*Lee et al., 1994*), as was the *flc-2* mutant in Col-FRI (*Michaels et al., 1999*). *FLC-Venus* and *FLC-mCherry* translational fusions were generated in a similar way to a previously published *FLC-GUS* reporter (*Bastow et al., 2004*). Specifically, either the *Venus* (*Nagai et al., 2002*) or *mCherry* (*Shaner et al., 2004*) coding sequence was inserted into the NheI site of *FLC* exon 6. The resultant 12.7-kb SacI/XhoI genomic fragments were transferred into pSLJ-755I6 (*Jones et al., 1992*) for transformation into *flc-2 FRI*-sf2 plants using *Agrobacterium tumefaciens*. Many (>30) independent transgenic lines that we analysed rescued the early-flowering phenotype of parental *flc-2 FRI*-sf2 plants and showed accelerated flowering in response to cold exposure. Transgenic *FLC* lines were selected for further analysis based on similarity between expression level of the transgene and that of endogenous *FLC*. Lines containing a single-copy transgene were identified using a qPCR-based assay adapted for *Arabidopsis* from a previously described method (*Bartlett et al., 2008*) (performed by IDna Genetics, UK). Primer and probe sequences are listed here: Bar-F: ggccgagtcgaccgtgta; Bar-R: ttgggcagcccgatga; Bar-Probe: FAM-cgccaccagcggacggga-TAMRA; AtCO-F: gtccgggtctgcgagtca; AtCO-R: gctgtgca-tagagaggcatcatc; AtCO-Probe: VIC-tgctccggctgctttttttgtgtgag-TAMRA. Single-copy *FLC-Venus* and

*FLC-mCherry* lines showed no evidence of transgene silencing in 3 generations of propagation after transformation.

## Expression analysis

Total RNA was prepared as previously described (*Etheridge et al., 1999*), except on a smaller scale. When measuring unspliced *FLC*, genomic DNA was removed using TURBO DNA-free (Ambion, Austin, TX, AM1907) following the manufacturer's guidelines, except that phenol–chloroform extraction and ethanol precipitation were used to further purify RNA after DNAse treatment. Reverse transcription was performed using the SuperScript III First-strand Synthesis System (Invitrogen, Austin, TX, 18080-051), according to the manufacturer's protocol using either gene-specific primers or Oligo(dT)12-18 (Invitrogen, 18418-012). *COOLAIR* isoforms were measured as previously described (*Csorba et al., 2014*). qPCR was performed using LightCycler 480 SYBR Green I Master (Roche, UK, 04887352001) on the LightCycler 480 instrument (Roche). Threshold cycle (Ct) values were calculated using the 'Second Derivative Maximum method' in the LightCycler software. RNA levels relative to *UBC* (At5g25760) (*Czechowski et al., 2005*) were determined using the ΔΔCt method (*Livak and Schmittgen, 2001*).

Primer and probe sequences are listed here: *UBC* (At5g25760, forward spans exon 3/exon 4, reverse in exon 4) UBCspl_F: ctgcgactcagggaatcttctaa; UBCspl_R: ttgtgccattgaattgaaccc; spliced *FLC* (forward spans exon 4/exon 5, reverse in exon 7) FLCspl_F: agccaagaagaccgaactca; FLCspl_R: tttgtccagcaggtgacatc; unspliced *FLC* (forward in intron 2, reverse in intron 3) FLC_3966_F: cgcaattttcatagcccttg; FLC_4135_R: ctttgtaatcaaaggtggagagc; *FLC-Venus* (forward in *Venus* coding sequence, reverse in *FLC* exon 7) FLC-VENUSex6_cDNA_1247_F: cacatggtcctgctggagtt; FLC-VENUSex6_cDNA_1388_R: cggagatttgtccagcaggt; Total *COOLAIR* (both primers in exon 1) Set6new_LP: tgtatgtgttcttcacttctgtcaa; Set6new_RP: gccgtaggcttcttcactgt; Class I *COOLAIR* (forward spans exon 1/exon 2, reverse in exon 2) Set2new_LP: tcatcatgtgggagcagaag; Set2new_RP: tctcacacgaataaggtggcta; Class II *COOLAIR* (reverse transcription primer and both qPCR primers in distal exon) Set4_RT: aatatctggcccgacgaag; Set4new_F-195: gtatctccggcgacttgaac; Set4new_R-195: ggatgcgtcacagagaacag.

## FLC-Venus pulldown and mass spectrometry

*FLC-Venus* or untransformed control plants were ground in liquid nitrogen and suspended in extraction buffer (20 mM Tris–HCl pH 7.5, 150 mM NaCl, 2.5 mM MgCl$_2$, 0.5% (wt/vol) Triton X-100, 10% (wt/vol) glycerol, cOmplete protease inhibitor EDTA-free [Roche, 04693159001]). After 10 min incubation with gentle rotation at 4°C, samples were cleared by repeated centrifugation at 20,000 × g, 4°C. Venus-tagged protein was precipitated by incubating soluble extract with GFP-Trap_M beads (Chromotek, Germany, gtm-20). Magnetic beads were washed three times with a mild wash buffer (20 mM Tris pH 8.0, 150 mM NaCl, 2 mM MgCl$_2$). Proteins were eluted by denaturation using sodium dodecyl sulfate (SDS), separated on polyacrylamide gels and either transferred to polyvinylidene difluoride membranes for analysis by immunoblotting or excised from gels for mass spectrometry. Liquid chromatography-MS/MS analysis was performed using a LTQ Orbitrap mass spectrometer (Thermo Fisher, UK) and a nanoflow-HPLC system (Surveyor; Thermo Fisher), as previously described (*De Lucia et al., 2008*). MS data were analysed using Scaffold 4 (Proteome Software, Portland, OR).

## Immunoblots

Whole seedlings were ground to a fine powder in liquid nitrogen and incubated for 10 min in lysis buffer (50 mM Tris pH 8.0, 100 mM NaCl, 5 mM ethylenediaminetetraacetic acid (EDTA), 1% (wt/vol) SDS, 5 mM β-mercaptoethanol, cOmplete protease inhibitor EDTA-free [Roche, 04693159001]). After clearing by centrifugation at 20,000 × g, 4°C, proteins were separated on SDS-polyacrylamide gels and transferred to nitrocellulose membranes (Hybond ECL; GE Healthcare, UK). Venus-tagged protein was detected with either a rabbit polyclonal anti-GFP antibody (Abcam, UK, ab290) or a commercial mouse monoclonal anti-GFP antibody mixture (Roche, 11814460001). Signals were visualized by chemiluminescence (SuperSignal West Femto; Pierce, Austin, TX) using secondary antibodies coupled to horseradish peroxidase (anti-mouse, Santa Cruz Biotechnologies, Dallas, TX; anti-rabbit, GE Healthcare). Membranes were reversibly stained using Ponceau S solution (Sigma–Aldrich, P7170).

## Confocal microscopy

Confocal microscopy data were obtained for homozygous single-copy FLC-Venus and FLC-mCherry lines at the T3 generation or F1 plants generated by crossing these lines. Imaging was performed using a 20×/0.7 NA multi-immersion lens, with water as the immersion fluid on a Leica TCS SP5 confocal microscope equipped with Leica HyD Hybrid detectors. For z-stacks, the step size was 3 μm, which meant that each nucleus was typically observed in 2–3 consecutive confocal z-slices. For single-fluorophore experiments with FLC-Venus lines, roots were immersed in 2 μg/mL propidium iodide (Sigma–Aldrich, P4864) to label the cell wall. The emission spectrum of propidium iodide overlaps with that of mCherry, so could not be used in FLC-mCherry or double-fluorophore experiments. The following wavelengths were used for fluorescence detection: FLC-Venus excitation 514 nm and detection 511–555 nm (with 514-nm notch filter), propidium iodide excitation 514 nm and detection 626–697 nm, FLC-mCherry excitation 561 nm and detection 570–620 nm. To allow comparison between treatments, the same laser power and detector settings were used for all FLC-Venus images and all FLC-mCherry images, respectively. For double-fluorophore experiments, Venus and mCherry fluorophores were simultaneously excited at 514 nm and 561 nm, respectively.

The following steps were used to prepare images for presentation: raw confocal z-stacks were aligned using the MultiStackReg plugin in Fiji (*Schindelin et al., 2012*). To reduce detector noise, a Gaussian blur with a 1.5-pixel radius was then applied to images measuring 2048 × 1024 pixels (510 × 255 μm), before taking maximum intensity projections over 2–4 z-planes (6–12 μm). Finally, the intensity was linearly adjusted separately for each channel. For visual comparison of nuclear intensity between different treatments in FLC-Venus images, the same linear adjustment was used.

## Quantitative image analysis

Confocal z-stack images were analysed using a custom image processing pipeline to reconstruct cellular volumes and calculate the mean FLC-Venus fluorescence intensity per cell. The pipeline is described below and summarized in *Figure 2—figure supplement 5*. The source code is available at https://github.com/JIC-CSB/root-image-analysis.

Since the propidium iodide cell wall stain is not compatible for imaging with mCherry, cell segmentation could not be performed for FLC-mCherry or double-fluorophore experiments. Image analysis was, therefore, undertaken for FLC-Venus only.

Pixels not corresponding to root tissue were masked using a series of morphological transforms of the propidium iodide (cell wall) images from each stack. To prepare the masked cell wall data for segmentation into individual cells, a Gaussian filter (using a standard deviation of 2 pixels) followed by median-based local thresholding (using Fiji's Auto Local Threshold plugin and a radius of 40 pixels) was applied to each plane of the stack. Each image was skeletonized using the Fiji Skeletonize plugin and then segmented using the Watershed plugin. Together, these steps generated an individual 2D segmentation for each cell wall image in the stack. The structure of the root was then reconstructed in 3D by comparing cells in segmented 2D images with those in neighbouring planes. Briefly, cells in 2D planes were considered part of the same 3D cell if the following two criteria were satisfied: first, their centroids were within a distance of 20 pixels from one another, and second, their relative areas did not vary by more than 50%. In addition to these criteria, the maximum extent of a single cell in the z-direction was limited to 18 μm (6 z-planes). This algorithm was implemented in Python (http://www.python.org) using the scikit-image library (http://scikit-image.org/). Finally, reconstructed 3D volumes were applied to the images from the FLC-Venus fluorescent channel to calculate mean intensity across the reconstructed volume by summing FLC-Venus intensity inside the reconstructed cell (from multiple z-planes) and dividing by the total volume of the reconstructed cell (summed area from multiple z-planes).

To validate the method, the mean FLC-Venus intensity per cell was estimated manually for a random selection of 50 cells from 22 different roots. Comparison of these results with those generated by the automated procedure for the same cells indicated that the mean cellular FLC-Venus intensities were accurate in approximately 80% of cells. The remaining cells in this test set were incorrectly segmented by the algorithm.

Consecutive z-stack images were separated by 3 µm and each root typically contained 14–18 images, which encompassed approximately the top third of the root in the meristematic and elongation zones. It was observed that FLC-Venus intensity decreased with depth in the image stack (*Figure 2—figure supplement 5B*). This effect may have arisen due to photobleaching as a greater depth also corresponded to a later image acquisition time. To reduce this effect, we restricted the analysis to those planes where the intensity was approximately constant. The number of roots analyzed and other statistics are shown in *Figure 2—figure supplement 5C*.

## Acknowledgements

We thank all members of the Dean and Howard groups for discussions. We also thank Nick Pullen and Charlotte Miller for comments on the manuscript and Grant Calder for technical assistance on the confocal microscope. This research was supported by an Advanced Investigator European Research Council grant MEXTIM, by grant BB/J004588/1 from the Biotechnology and Biological Sciences Research Council and by The John Innes Foundation.

## Additional information

### Funding

| Funder | Grant reference | Author |
| --- | --- | --- |
| Biotechnology and Biological Sciences Research Council | BB/J004588/1 | Scott Berry, Matthew Hartley, Tjelvar S G Olsson, Caroline Dean, Martin Howard |
| European Research Council | MEXTIM | Caroline Dean, Martin Howard |
| John Innes Foundation | Rotation PhD Studentship | Scott Berry |

The funders had no role in study design, data collection and interpretation, or the decision to submit the work for publication.

### Author contributions

SB, Conception and design, Acquisition of data, Analysis and interpretation of data, Drafting or revising the article; MH, TSGO, Computational image analysis; CD, MH, Conception and design, Analysis and interpretation of data, Drafting or revising the article

### Author ORCIDs

Scott Berry, http://orcid.org/0000-0002-1838-4976
Matthew Hartley, http://orcid.org/0000-0001-6178-2884
Tjelvar S G Olsson, http://orcid.org/0000-0001-8791-4531

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
