## [Decision Letter]

Thank you for sending your work entitled “Local chromatin environment of a Polycomb target gene instructs its own epigenetic inheritance” for consideration at *eLife*. Your article has been evaluated by Naama Barkai (Senior editor), a Reviewing editor, and two reviewers. The overall consensus was that, although the inheritance of H3K27 methylation in cis is largely a settled matter, the results reported in your paper are important to the Polycomb-Group field and the general field of epigenetics because they provide the most compelling data to date that PcG-mediated gene silencing operates in cis.

The Reviewing editor and the reviewers discussed their comments before we reached this decision, and the Reviewing editor has assembled the following comments to help you prepare a revised submission.

1) It's important to know how *COOLAIR* is expressed in the lines with transgenes. Does the insertion of the reporter disrupt expression? If *COOLAIR* is not expressed normally in these cells, some comment upon how that reflects its role in regulation is warranted.

2) Because the two transgenes are not equivalent (e.g. in different genomic locations, have different insertions), there is a possibility that differential sensitivity of the transgenes to diffusible PRC2 accounts for the results. Although such a possibility appears remote, it should be discussed more thoroughly.

3) Please rephrase the paragraph of the Introduction that discusses silencing at the yeast GAL1 locus. The implication that the cited reference (Kundu, 2007) concludes that epigenetic memory at GAL1 is stored in cis may be misleading, because the authors of that paper (Kundu et al.) found no effect of local chromatin state on re-induction kinetics of GAL. The authors did find that that nucleosome remodeling factors are necessary for memory at GAL, but that does not mean it happens in cis.

---

## [Author Response]

*1) It's important to know how* COOLAIR *is expressed in the lines with transgenes. Does the insertion of the reporter disrupt expression? If* COOLAIR *is not expressed normally in these cells, some comment upon how that reflects its role in regulation is warranted.*

*FLC-GUS* and *FLC-LUCIFERASE* constructs have previously been generated in our lab using a similar cloning strategy to that employed for *FLC-Venus* and *FLC-mCherry* ([4]; Mylne et al., 2004). It is important to note that the insertion site for the reporter coding sequence (*GUS, LUC, Venus, mCherry*) is in exon 6 of the *FLC* mRNA, rather than at the 3′ end. This means that the inserted sequence lies downstream of the polyadenylation site for the Class I *COOLAIR* isoform (in the direction of antisense transcription) and within the large intron of the Class II *COOLAIR* isoform. The previously identified splicing and polyadenylation sites of *COOLAIR* are therefore not perturbed in the transgenes.

Following the reviewers' request, we measured *COOLAIR* expression in the *FLC-Venus* and *FLC-mCherry* lines. We found that both Class I and Class II *COOLAIR* are expressed from the transgenes. For the different isoforms, we found that the quantitative levels of non-vernalized *COOLAIR* expression varied up to approximately 2-fold between transgenes and endogenous *FLC*, although we did not detect any consistent pattern of differences between transgenic lines and endogenous *FLC*. We found that *COOLAIR* expression is greatly increased during cold exposure (predominantly Class I) in *FLC-Venus* lines, as previously shown for endogenous *FLC* (45; 9). We have added these results to a new figure supplement (Figure 2—figure supplement 2), which also shows a schematic of the *COOLAIR* splicing isoforms detected and their relationship to the Venus/mCherry insertion site.

The role of *COOLAIR* in vernalization is currently unclear. Plants carrying insertions in the *COOLAIR* promoter, and reduced *COOLAIR* expression, still show stable epigenetic silencing of *FLC* after cold exposure ([Bibr bib23a]). Recently, transgenic *FLC* constructs lacking the *COOLAIR* promoter were used to demonstrate a role for *COOLAIR* in promoting transcriptional down-regulation of *FLC* during cold exposure (9). However, accumulation of H3K27me3 at *FLC* during cold was not affected by removal of *COOLAIR*. Since the role of *COOLAIR* in vernalization is currently unclear, and *COOLAIR* expression does not show informative differences between transgenic lines and endogenous *FLC*, we have maintained a minimal discussion of *COOLAIR* in our main text. We have added the following sentence to the end of paragraph 1 of the Results: ‘We also verified that the *FLC* antisense transcripts, named *COOLAIR* (45; 9), are expressed from these constructs and are induced during cold exposure (Figure 2—figure supplement 2).”

We have also added RT-qPCR primer sequences and a reference to how *COOLAIR* was measured to the Materials and methods section “Expression analysis”.

*2) Because the two transgenes are not equivalent (e.g. in different genomic locations, have different insertions), there is a possibility that differential sensitivity of the transgenes to diffusible PRC2 accounts for the results. Although such a possibility appears remote, it should be discussed more thoroughly*.

If there were substantially different sensitivity in that rate of silencing of the two transgenes due to genomic location differences, this should be evident in differences in flowering time after vernalization, which is quantitatively related to *FLC* epigenetic repression. Such differences are not observed in our data (Figure 1). Furthermore, if this were the case, we would expect one of the transgenes (either *FLC-Venus* or *FLC-mCherry*) to be preferentially silenced before the other in F1 plants. That we see no obvious bias in the amount of FLC-Venus/FLC-mCherry ON/OFF versus OFF/ON cells, suggests that both transgenes have similar probabilities of being repressed during cold. To explain this more thoroughly we have added the following sentence to the Results and Discussion section: “Furthermore, since we observe similar numbers of cells in the *FLC-Venus/FLC-mCherry* ON/OFF and OFF/ON expression states (Figure 3—figure supplement 2), we consider it unlikely that differences in genomic location or fluorophore sequence between the transgenes causes one of the copies to be preferentially repressed over the other.”

To minimise the effect of the surrounding genes on transgenic *FLC* regulation, a large section of genomic DNA surrounding *FLC* was used in the transgenic construct, including the closest downstream gene and part of the upstream gene. Many (>30) independent transgenic lines that we analysed showed complementation of the early-flowering phenotype and accelerated flowering in response to cold exposure. We detected quantitative expression level differences between lines and selected those with similar levels to endogenous *FLC*. To describe this in more detail, we have added the following sentences to the Materials and methods: “Many (>30) independent transgenic lines that we analysed rescued the early-flowering phenotype of parental *flc-2 FRI-sf2* plants and showed accelerated flowering in response to cold exposure. Transgenic *FLC* lines were selected for further analysis based on similarity between expression level of the transgene and that of endogenous *FLC*.”

The insertion site of the fluorophore sequence is towards the 3′ end of the *FLC* sequence. This region has not previously been shown to be required for epigenetic repression. Important regulatory sequences for cold-induced epigenetic repression that have been identified are located in intron 1 ([Bibr bib43a]; [Bibr bib23a]; [Bibr bib7a]), close to the small “nucleation region” in which H3K27me3 arises during cold exposure (1). Using insertions of similar length and sequence, we have minimised differences between the two transgenes while maintaining the ability to discriminate the two copies.

*3) Please rephrase the paragraph of the Introduction that discusses silencing at the yeast GAL1 locus. The implication that the cited reference (Kundu, 2007) concludes that epigenetic memory at GAL1 is stored in cis may be misleading, because the authors of that paper (Kundu et al.) found no effect of local chromatin state on re-induction kinetics of GAL. The authors did find that that nucleosome remodeling factors are necessary for memory at GAL, but that does not mean it happens in cis*.

We agree that the way this paragraph was constructed and referenced in the original submission was potentially confusing. The intention of including this was to provide a concrete example of where a chromatin-modifying factor is required for epigenetic memory, but where memory is actually stored in trans. Since this section was obviously not helpful in conveying this message, and is not essential for the remainder of the paper we have removed all specific discussion of the GAL locus. In this section, we chose to maintain the statement “importantly, a requirement for chromatin-modifying factors in epigenetic memory does not necessarily imply that memory is itself stored in chromatin” without reference. This now appears at the end of the third paragraph of the Introduction.